# News about the Role of Fluid and Imaging Biomarkers in Neurodegenerative Diseases

**DOI:** 10.3390/biomedicines9030252

**Published:** 2021-03-04

**Authors:** Jacopo Meldolesi

**Affiliations:** Division of Neuroscience, San Raffaele Institute and Vita-Salute San Raffaele University, via Olgettina 58, 20132 Milan, Italy; meldolesi.jacopo@hsr.it

**Keywords:** neurons, astrocytes, Alzheimer’s and Parkinson’s diseases, fluid and imaging biomarkers, amyloid-β and tau, miRNA, extracellular vesicles, exosomes and ectosomes, PET, radiotracers, radiolabeled molecules

## Abstract

Biomarkers are molecules that are variable in their origin, nature, and mechanism of action; they are of great relevance in biology and also in medicine because of their specific connection with a single or several diseases. Biomarkers are of two types, which in some cases are operative with each other. Fluid biomarkers, started around 2000, are generated in fluid from specific proteins/peptides and miRNAs accumulated within two extracellular fluids, either the central spinal fluid or blood plasma. The switch of these proteins/peptides and miRNAs, from free to segregated within extracellular vesicles, has induced certain advantages including higher levels within fluids and lower operative expenses. Imaging biomarkers, started around 2004, are identified in vivo upon their binding by radiolabeled molecules subsequently revealed in the brain by positron emission tomography and/or other imaging techniques. A positive point for the latter approach is the quantitation of results, but expenses are much higher. At present, both types of biomarker are being extensively employed to study Alzheimer’s and other neurodegenerative diseases, investigated from the presymptomatic to mature stages. In conclusion, biomarkers have revolutionized scientific and medical research and practice. Diagnosis, which is often inadequate when based on medical criteria only, has been recently improved by the multiplicity and specificity of biomarkers. Analogous results have been obtained for prognosis. In contrast, improvement of therapy has been limited or fully absent, especially for Alzheimer’s in which progress has been inadequate. An urgent need at hand is therefore the progress of a new drug trial design together with patient management in clinical practice.

## 1. Introduction

Biomarkers are molecules that are highly variable in their origin, nature, and mechanism of action, and they are connected to or are directly involved to single or various peculiar diseases. At present, biomarkers, which are addressed to cells, organs, or structures, exist for almost all diseases, including cancers. Among them, neurodegenerative diseases are receiving the greatest attention. During the last 20 years, articles about their biomarkers published in known journals have totaled over 20,000, including about 4000 reviews. Investigations of their properties, going from specificity to the mechanisms of their action, are often used to clarify various aspects of pathogenesis. Such results often play roles in processes of medical relevance, such as diagnosis, prognosis, and also therapy, and they are useful for patients and also for clinical practice. However, the relevance of biomarkers in clinical practice is variable. Some of them are well known and widely used; for others, however, knowledge is still questioned due to, for example, their limited specificity. In this review such limitations are not further illustrated. The properties of biomarkers presented here are those of general significance, identified in the last few years. Information about additional aspects can be found in other publications [1,2].

Compared to biomarkers of other organs, biomarkers of the brain exhibit distinct properties [3,4]. Initial studies about two types, the fluid and the imaging biomarkers, were developed separately. Biomarkers of the first type appeared around 2000. They are collected not in vivo but in fluid, within either of two fluids taken from patients, the central spinal fluid (CSF) and the blood plasma [5,6]. Biomarkers of the second type were developed by the use of radiolabeled molecules. Upon penetration into the living brain, these molecules are bound with high specificity as revealed by positron emission tomography (PET) imaging. A biomarker study by the latter approach, addressed to amyloid-β (Aβ) plaques of Alzheimer’s disease (AD), was published in 2004 [7]. Since then, the in vivo studies of the imaging type have continued with growing success (see Section 5). Since the beginning of the two types of biomarker studies, the state of neurodegenerative diseases changed profoundly. In particular, biomarkers did recently revolutionize scientific and medical research by transforming drug trial design and also improving patient management in clinical practice [8]. 

So far, I have introduced the two types, i.e., fluid and imaging biomarkers. The in vivo imaging of the latter takes place upon their high affinity binding by radiolabeled specific molecules introduced in the brain. In contrast, fluid biomarkers are collected within (CSF) or away (blood) from the central nervous system. Yet, all fluid biomarkers are adequate to identify central molecules/processes, critical for patients suffering neurodegenerative diseases, not only AD but also Parkinson’s disease (PD), amyotrophic lateral sclerosis (ALS), and other diseases. Initially, the disease identification was searched by the recovery in the fluids of free specific molecules. These molecules, however, are largely digested during their traffic, from the cells affected by the disease to the accumulation in the fluids. Thus, their identification was often difficult. The problem has been solved recently by changing the study from free molecules to cargo molecules segregated within extracellular vesicles (EVs). This change will be presented in detail in Section 3.

Initially, each neurodegenerative disease was considered dependent on a single specific biomarker, such as Aβ for AD and α-synuclein for PD. Now it is clear that these two, as well as many other biomarkers, are not fully specific but expressed also by patients of other neurodegenerative diseases [9]. Multispecificity of biomarkers requires caution in their operation. Caution is necessary also with another type of problem. Initially considered specific and efficient, some biomarkers have been found to be hardly reproducible, thus inappropriate for research and clinical practice [8]. A final important consideration refers to the age dependence of disease investigation/treatment. Specific biomarkers have been applied not only to mature patients, but also to patients at early stages of disease. By novel and innovated methods, it could be established whether patients at risk of long-term diseases, such as AD or dementia, can be treated only upon full development of their symptoms, or also when symptoms are absent or still at an early stage [10,11]. 

Summing up, I anticipate the two types of biomarker generation, and the anticipation of their properties confirms their relevance, which is demonstrated for many of them. The others, in contrast, are weak, characterized by poor specificity and limited employment. The review here includes the most important properties of the two types of biomarkers that have been identified during the last years. Following this introduction, Section 2 deals with the fluids, the environment of type one biomarker generation, followed by Section 3 about the role of EVs, and Section 4 about fluid biomarkers. Finally, Section 5 deals with the second type of biomarkers, i.e., the imaging biomarkers.

## 2. Fluids

In the Introduction, I already mentioned the two ways leading to the generation of neural biomarkers, i.e., the first based on the analysis of peripheral fluids containing appropriate molecules, the second in vivo, based on the imaging by PET labeling of specific brain molecules. How is it that fluids are essential for the first, very important approach? Two of their properties need to be emphasized. First, molecules of interest for their recognition as biomarkers need to be present in the biological fluids of human body; second, the analysis of fluids, with ensuing isolation of molecules, is less expensive, much easier, and does not disrupt the body compared to the in vivo analysis of tissues. In other words, the fluids are advantageous in the development of biomarkers.

In the body of mammal animals, including humans, there are 8 types of external fluids. For our purposes, however, only two are relevant: the CSF and the blood plasma. Molecules and the small organelles EVs, released from brain cells such as neurons and astrocytes, navigate in the extracellular fluid space from which they are easily transferred to the CSF [3] (step 2 in Figure 1). At the arachnoid villi, the EVs of the CSF can move to venous blood within large vacuoles (step 4 in Figure 1). In addition, the molecules and EVs have been shown to traffic through the blood–brain barrier [12,13] (step 3 in Figure 1). The latter is the structure known to reduce/exclude the traffic of many other molecules and organelles to and from the brain. 

Recent studies have shown that biomarkers of the two fluids, CSF and blood plasma, are employed approximately with the same frequency. The choice of one fluid, however, is not due to a negative evaluation of the other. In recent reviews, both are reported as valid [12,14]. Rather, the choice appears to depend on advantages existing in either fluid. High levels of molecules and EVs are present in CSF (Figure 1), which is more invasive and requires more expenses than the blood for biomarker generation. Yet, the CSF fluid has made it possible to have processes of low concentrated proteins, for example, those of the COVID-19 disease reported this year in patients exhibiting neurological symptoms, evident of cerebral infection [15]. In the blood, plasma withdrawal can be large, frequently repeated, and not expensive. However the levels of molecules/EVs in this fluid are lower than in the CSF. The transfer of various EVs across the blood–brain barrier (BBB) has been shown to be different, with ensuing variability of the plasma levels [16]. In many cases, however, they have emerged in the study of neurodegenerative diseases, with enormous potential as a diagnostic, evaluation of therapeutics, and treatment tool. 

Summing up, both fluids are employed for biomarker generation. However, the choices depend on their distinct properties. The processes of biomarker generation, previously defined of liquid biopsy [13], have been shown to identify not only the fluid molecules but also the brain pathologic EVs inaccessible in vivo. In addition, fluids have offered unique opportunities, as seen from recent clinical trials, to improve the quality and applicability of results. The processes involved will be developed by approaches presented in the subsequent Section 4 of this review.

## 3. From Molecules to Extracellular Vesicles

Neurodegenerative diseases are heterogeneous disorders characterized by a progressive and severe cognitive and functional decline leading to the progressive loss of function and death of neuronal cells. Due to the complexity of their diagnosis, early detection and treatments are difficult to recognize, and this is critical for the development of successful therapies. In addition, current diagnostic approaches are often poorly effective. Thus, their therapeutic effects are limited [16]. For several years, progress in diagnosis and therapy has been searched by molecules, such as proteins and nucleotides, via the generation of biomarkers, specific in the diagnosis and possibly in the therapy of the diseases investigated. However, during their traffic and maintenance within the fluids, the molecules were extensively digested by proteases and ribonucleases. Thus, the attempts based on free proteins and nucleotides remained with limited success. The attempt changed considerably when the search for biomarkers started to be made by the use of EVs containing the molecules of interest within their lumen (Figure 1). Now it is clear that when segregated within EVs, the molecules are protected, making them suitable candidates for noninvasive biomarkers [16,17,18]. 

EVs are small vesicles of two types released by all cells. The first, widely called exosomes, are produced and accumulated within a vacuole of endosomal nature, the multivesicular body (MVB); the second, the ectosomes, are released by shedding of rafts, directly from the plasma membrane (Figure 1). Interestingly, the two EV types share similar properties. Their membranes are different from the two membranes of origin; their cargoes, segregated within their lumen, contain peculiar molecules, namely many proteins, many nucleotides, most often the noncoding miRNAs, lipids, and others. EVs of neuronal and astrocytic origin, when harvested from blood, can be used to interrogate brain pathologic processes. Moreover, in the case of proteins involved in the development of neurodegeneration, their segregation within EVs and even more their transfer to the fluids decrease the levels within the cells and can thus alter the disease progression [17]. 

Isolation from CSF or blood plasma of specific brain-cell-derived EVs, confirmed by neural markers [12], is theoretically simple. Thus, the approach has been successful recently and offers great promises for the near future [17,18]. The EVs, released by neurons and astrocytes of patients affected by neurodegenerative disease, can be identified by specific biomarkers found in their cargoes, such as Aβ and phosphorylated tau in AD, α-synuclein in PD, and the transactive response DNA/RNA binding protein of 43kDa (TDP-43) in ALS [13,19,20,21,22] (Table 1). Other biomarkers investigate the subclinical declines of the diseases, for example, the decline of middle age in AD. This can be done with tau and insulin signaling biomarkers [23,24]. A problem considered is that of multiple dementias, only some are connected to neurodegenerative diseases such as AD and PD. Among the proteins of EVs from specific neurodegenerative diseases, some have been recently shown to concern synaptic and axon injury, inflammations, stress responses and other defects [14,25,26,27,28]. Other dementias that are independent of neurodegenerative diseases still need identification of their specific biomarkers [29].

In addition to many proteins, EVs include in their cargo various types of nucleotides, with predominance of miRNAs. The latter are members of a noncoding family involved in various functions, the best known being translational gene expression. Within fluids, miRNAs investigated have been numerous. Whether or not they play the role of biomarkers has been discussed. Positive evidence, analogous to that of proteins, has been recently reported but without precise identification of many miRNAs involved [30,31,32,33]. Interestingly, biomarkers of miRNA origin were shown active in the ALS disease [31]. Another positive conclusion has been found, but it concerns children, healthy and patients of their diseases [33]. Additional studies have focused on a new form of noncoding RNA, namely circular RNA (circRNA), known to traffic within EVs. At present, however, the function of these RNAs is unknown. CircRNAs might be involved in age-related diseases [34,35]. RNAs of this type may also operate in neuropsychiatric disorders [35]. The next section, focused exclusively on AD, intends to reconsider in more detail the origin, the multiplicity, and the role of its biomarkers, with special attention on their heterogeneity with perspectives of innovative therapies.

## 4. AD and Its Multiple Fluid Biomarkers

AD is the neurodegenerative disease of greatest importance for two reasons, namely its much larger number of patients (at least 50 million worldwide) and the many scientists committed to its investigation. The choice of this disease has been made to illustrate its general properties. The task is to provide a current landscape, largely common to PD [21,29,36] but not always to other neurodegenerative diseases (see, for example, [9,17,28,31,32]). 

A property common to many neurodegenerative diseases is heterogeneity. Various factors are considered as possible con-causes of its starting. Among these are oxidative stress, neural network dysfunctions, and defects in protein regulation and degradation. In the case of AD, two additional factors need to be considered, i.e., inflammation and immune dysregulation [37]. Such defects are expected to induce alterations in neurons, synapses, axons, and possibly also on glial cells. As already assumed in Section 3, some of these factors have been recognized by the identification of the corresponding biomarkers. For other defects, however, biomarkers are not available yet. Although incomplete, the present knowledge appears of interest to establish the properties of some AD heterogeneous forms, such as responses to treatments. The study of multiple biomarkers, identified during the last months, might be sufficient to characterize at clinical level various aspects of AD, including diagnosis and prognosis [38,39,40]. Heterogeneity exists also for tau. The fraction inducing tau biomarkers in the CSF was found to stage Alzheimer’s disease, and this is potentially useful for tau-directed therapeutics [41]. Other proteins, such as mesenchymal stem cells and exosomes, appear of considerable potential for therapy. The role of some other properties, including heterogeneity, remains to be established [42,43]. 

An additional approach relevant for AD diagnosis and therapy has been recently reported based on two different miRNAs that are active as EV biomarkers in the blood plasma. Although not identical, the evaluation of the two miRNAs could be considered in parallel. Effects in AD patients, induced by increased miRNA levels in the blood, were revealed by neuronal viability and neuroinflammation followed by a mini-mental state examination. Additional effects were investigated also in fluid, using well-known neural cells such as SH-SY5Y cells, affected by Aβ treatment and evaluated in terms of proliferation, apoptosis, and neuroinflammation. Interestingly, significant upregulation of the first miRNA, miR-485-3p, was shown in patients and cell models, accompanied by severity of DA in vivo and in fluid [44]. The response induced by the second miRNA, miR-331-3p, appears the opposite. Both in vivo and in fluid, the increased miRNA induced significant and persistent attenuation of AD [45]. The two types of results induced by these miRNA opened the possibility of new, promising therapies based on the two biomarkers investigated [44,45]. 

In conclusion, the actions by EV biomarkers based on multiple properties of proteins and miRNAs have recently offered ample chances of both nature and specificity, showing enormous potential as diagnostic, evaluation and treatment tools. Their new results allow researchers to test hypotheses by proof of concept studies at the preclinical phase, with further opportunities to develop therapeutic discoveries in neurodegenerative diseases [46]. 

## 5. Imaging Biomarkers from AD and PD

As already reported in Section 1, the identification of Aβ as the key factor of AD, which was already demonstrated in fluid, was confirmed in vivo by the development and investigation of an appropriate radiolabeled molecule. Upon its penetration into the living brain, such a molecule made it possible to have a clear Aβ imaging by PET [7]. By now, such an approach of investigation plays roles much more important than its historical role. Advanced technologies of the near future are expected to further improve the present success of such studies. The biomarkers obtained by this approach are usually named imaging biomarkers. 

The advantages offered by such biomarkers include properties that are different and integrative with respect to the fluid biomarkers illustrated in the previous sections. Among the results of imaging biomarkers are their demonstrations occurring in the living brain, distinct from the peripheral demonstrations of fluid biomarkers; their data obtained from larger numbers (hundreds) of patients; and the quantitation of their measurements, necessary for many assays. The relevance of these properties emerges from a recent group of papers based on the comparison between the results obtained with fluid and imaging biomarkers. In these reports, the relevance of the results obtained by the imaging techniques is emphasized [36,47,48]. Another difference between the two approaches has emerged with respect to the amyloid cascade, a concept activated by elevated levels of Aβ and tau, which is completed by severe cognition and functional impairments [49]. In this case, the data by CSF biomarkers were found to emerge before those by PET imaging, demonstrating a high sensitivity that is relevant for the study of early AD stages [50]. Further studies with CSF biomarkers, compared with those obtained by PET and resonance (MRI) imaging, confirmed the relevance of the latter approach in many preclinical AD investigations [51,52]. 

Among additional problems adequately investigated by brain imaging is the heterogeneity of neurodegenerative diseases. Detailed recent studies have focused not only on the various forms of AD but also on PD and its atypical syndromes. In an initial study, several PDs that were hindered by substantial clinical and pathological heterogeneities were investigated by numerous imaging techniques including PET and MRI [53]. More recently, combinations with new techniques have been established to investigate, in distinct areas and pathways of the brain, the various molecules and processes, i.e., the molecular imaging of Aβ, tau and α-synuclein, as well as neuroinflammation. This integrative “multimodal approach” was found to be superior to single modality-based methods, with expected future advancements in the field [54]. Other innovative combinations of imaging biomarkers, revealed by MRI and PET, have been employed to investigate subtypes of AD. At the level of previously established groups of patients, the average and clinical characteristics appeared similar. In subtype assignments, however, disagreements were considerable. The subtypes therefore need to be further investigated, with an establishment of their harmonization by appropriate methods [55]. 

Additional new studies have expanded the investigation of in vivo biomarker-driven profiles by the reinforcement of the techniques employed. In a first example, a versatile form of PET imaging was found able to quantify the molecular targets of interest. By such approach, age-related neurodegenerative diseases, inducing dysregulation of synapses, neuroinflammation, protein misfolding, and other dysfunctions, were appropriately identified. Discussion of these processes has led to the identification of novel biomarkers [56]. A second approach was based on the development of optical imaging (OPI) probes and devices, which are affordable by imaging studies but are limited by their low depth of penetration. The combination of the OPI technique with PET, which is characterized by high depth penetration, resulted in the elimination of each limitation. The affording of new radiolabeled fluorophores made the activation of the dual PET/OPI mode possible, with excellent preclinical imaging results in various pathological conditions [57]. 

Summing up, the imaging biomarkers presented in this section are profoundly different from the biomarkers of the fluid type. Being recognized from the analysis of brain tissue, they do not fit strictly the definition of biomarkers as distinct molecules related to pathology, given in the first lines of the Introduction. Yet, a growing class of radiotracers, addressed to specific proteins such as Aβ and tau, have recently contributed to the growing knowledge about AD. In other words, the targets of the imaging processes can now be identified as molecules based on which the identification of imaging biomarkers is established [58,59]. 

## 6. Conclusions

This review focused on the new developments of two types of biomarkers, i.e., fluid and imagining biomarkers, generated via two distinct operative pathways originated in the brain. As emphasized in Section 5, even if the structure and the mechanisms of action of the two types are largely different, at least some of their effects are similar, to the point that their results, when considered together, give rise to positive integrations [36,47,48]. 

How relevant is the medical work of biomarkers? The experience accumulated during the last decade appears positive and encouraging. Many of their properties, such as specificity, multiplicity, and affinity, have strengthened their use in the course of AD, PD, and other neurodegenerative diseases. The present investigation about the pathogenesis of AD and the other neurodegenerative diseases is largely due to biomarker-based data. To understand the issues underlying complex symptoms such as dementia, biomarkers often operate combined with other disciplines [60]. 

In basic and applied research, the impact of biomarkers has increased, sustained by the improvement of their technologies. In the fluid field, the recent experience with EVs as tools for biomarker generation has improved their success considerably. An additional development is the recent recognition of increased number of proteins and peptides in the CSF of AD patients [61]. The imaging technology has been strengthened by the development of new radiolabeled molecules, by the use of improved imaging techniques, and by their combination with different procedures. In these studies, the best successes has been obtained when working on early stages of AD, including the identification of pathological alterations and a selective cognitive decline [62]. 

The results obtained by the use of biomarkers are often of relevance in medical practice. This in particular is the case of diagnosis: O often inadequate when based only on medical practice, it has been recently improved by the multiplicity and specificity of biomarkers. By the use of the latter, it has been established that a disease can be diagnosed even before the appearance of symptoms [10,11,51,52]. Moreover, the intensity of the responses can contribute to the prognosis of patients [28,44]. Thus, diagnostic and prognostic results obtained by biomarkers are often analogous. In contrast, therapy is still problematic. Although open to all neurodegenerative diseases, such problem is particularly aggressive for AD and ALS, for which appropriate therapy is not available. An initial attempt to improve the present situation involves the investigation of diseases at early or even pre-symptomatic conditions. Drugs slowing down the development of the disease should delay its progress, ultimately resulting in a progressive decrease of the AD mature state. Up to now, however, the results of clinical trials to evaluate the state of the AD therapy have been disappointing. In the future, development should be based on multiple criteria, including the state of the disease, its progression, and the activity of biomarkers focused on critical processes [40].

In conclusion, the recent identification and employment of biomarkers have resulted in increased knowledge and improvement of both basic and applied research. However, as emphasized in the first and subsequent sections of this review, various aspects of their function are still not completely or even inappropriately known. In the future, it appears desirable that intense research about biomarkers will include their critical evaluation and their further improvement, especially in terms of new therapy practice.

## Figures and Tables

**Figure 1 biomedicines-09-00252-f001:**
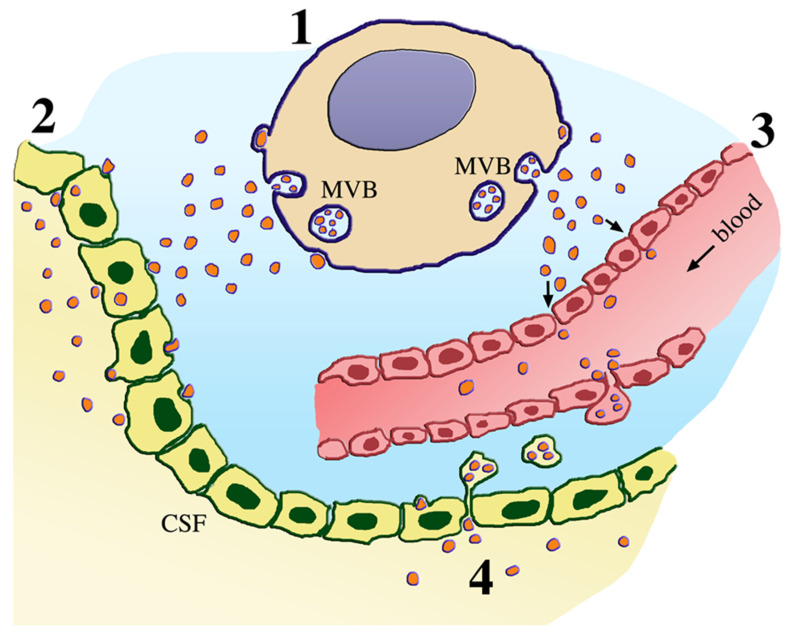
Traffic of extracellular vesicles (EVs) released by a neural cell. The neural cell at the top (cytoplasm of grain color, marked 1) releases the two types of EVs, the small exosomes by exocytosis of multivesicular bodies (MVB), and the larger ectosomes by shedding of plasma membrane rafts. Upon release, the vesicles navigate in the extracellular fluid (light blue). Their targeting relevant for fluid biomarker generation can be accumulate either to the central spinal fluid (CSF) of the ventricular system (left, peach color, marked 2) or as shown by the arrows to the blood (right, red-pink color, marked 3). Additional CSF-to-blood EV transfer occurs by large vacuoles operative at the arachnoid villi (bottom, marked 4). Thus, molecules and EVs can move from the extracellular space and the CSF to the blood plasma. In the case of neurodegenerative diseases, the molecules and EVs of such origin account for significant fractions of the total transferred to the fluids.

**Table 1 biomedicines-09-00252-t001:** Gene expression and disease specificity of proteins/biomarkers.

Genes	Proteins/Biomarkers	Diseases
*APP*	amyloid precursor protein/Aβ *	AD
*PSEN1*	presenilin1 *	AD
*PSEN2*	presenilin2	AD
*MAPT*	tau *	AD, PD, DLB, FTD
*C9orf72*	C9orf72 *	FTD, ALS
*GRN*	progranulin *	FTD, ALS
*VCP*	valosin-containing protein *	ALS, FTD, PD
*TARDBP*	TDP-43 *	ALS
*FUS*	fused in sarcoma (FUS)	ALS
*HTT*	huntingtin *	HD
*SNCA*	α-synuclein *	PD, DLB, AD
*GBA*	β-glucocerebrosidase	PD, DLB
*ApoE*	apolipoprotein-E	AD (risk factor)
*TREM2*	TREM2	AD (risk factor)

Well-known fluid biomarkers related to the mentioned proteins are marked by an asterisk (*). The dependence of several diseases may explain the low specificity of some fluid biomarkers. Abbreviations not used elsewhere in this Review: C9orf72, chromosome9 open reading frame 72 gene; DLB, dementia with Lewy’s bodies; FTD, frontotemporal dementia; FUS, fused in sarcoma gene; HD, Huntington disease; MAPT, gene of the microtubule-associated protein tau; SNCA, synuclein alpha gene; TARDPB, TAR DNA binding protein gene; TREM2, triggering receptor expressed on myeloid cells 2.

## Data Availability

All the data reported in this review are available at the American Program PubMed.

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
