# Peer review of "News about the Role of Fluid and Imaging Biomarkers in Neurodegenerative Diseases"

_biomedicines, 2021, doi:10.3390/biomedicines9030252_

Round 1
Reviewer 1 Report
The manuscript describes the biomarkers for neurodegenerative disease present in biological samples. The biomarkers are of two types - one found in body fluids and the imaging biomarkers. in the paper the author focuses not only on biomarkers used in the diagnostics of Alzheimer and Parkinson disease, but discusses the importance of certain body fluid for diagnostics..
major
The paper will benefit if clinical validity of biomarkers will be presented in particular
- in which stage of the disease or even before the onset of clinical signs the biomarkers can be detected;
- if it is possible to determine the severity of the disease by analysing the measured values of biomarkers;
-if the measured values biomarkers affect the progression of disease and or do they have prognostic value.
Minor points.
- line 77 α( alpha)-synuclein not a-synuclein
- line 133 the sentence is not finished
- line 260 the sentence is not finished
Author Response
This reviewer was favorably impressed by the review. She/he decided however a major criticism concerning the insufficient demonstration of the clinical validity of biomarkers. For this reason the Reviewer recommended the inclusion of three clinical properties in which biomarkers are relevant:
- the early diagnosis of diseases
- the severity of diseases
- the prognostic value
These clinical relevance effects of biomarkers were already reported in the review, however separately, at some distance from one another. I have therefore reconsidered in sequence these properties of biomarkers in the Conclusion, the final section of the review, specifying their relation to the clinical relevance. In my opinion such presentation in the Conclusion answers the major request of the reviewer.
Minor points
I have corrected a-synuclein into α-synuclein.
No absence of sentences appear in the official text I received.
Reviewer 2 Report
The review article by Jacopo Meldolesi is well organized and written and focuses on describing the current state of using biomarkers in neurodegenerative disorders such as Alzheimer's and Parkinson's disorders. I have a few relatively minor comments.
- My first comment to the author is that the review would benefit from describing the criteria by which a given biomolecule is defined as a biomarker. Not any biomolecule can be considered as a biomarker, and the reader would benefit from understanding what makes a given molecule a biomarker. I understand that the author specifically talks about neurodegenerative disorders that have been extensively studied, and as therefore, all of the fluid biomarkers have been extensively validated as such.
- Generally in vitro would apply to molecules generated outside of a living organism, e.g., in cell culture. To that extent, I'd say that biomolecules isolated from CSF or blood plasma would still be considered in vivo biomarkers.
- It would be beneficial if the author has also discussed the approaches by which the EVs from neurons or astrocytes are differentiated.
Author Response
This reviewer emphasized my review to be well organized and focused on biomarkers of neurodegenerative disorders. She/he had only a few, relatively minor comments to emphasize.
- Definition of biomarkers. This exists already in the Introduction of the review. Specifically it is emphasized our interest to be in the specific biomarkers of neurodegenerative diseases. For the details that are left out the readers are addressed to other comprehensive reviews already present in the literature.
- Biomarkers in fluid cannot be defined in vitro since they are taken from spaces that belong to living animals. I agree that the definition in vitro was not correct. On the other hand, its definition needed to be different from that of imaging biomarkers, entirely in vivo in this case. I have therefore changed in vitro into in fluid, certainly the environment of the EVs and their biomarkers when taken up.
- Distinction of EVs generated from AD and PD. This was already specified in the text, dependent on the two specific markers. I do not think necessary re-define this mechanism.